# Analysis of the Conformational Landscape of the N-Domains of the AAA ATPase p97: Disentangling the Continuous Conformational Variability in Partially Symmetrical Complexes

**DOI:** 10.3390/ijms25063371

**Published:** 2024-03-16

**Authors:** Sepideh Valimehr, Rémi Vuillemot, Mohsen Kazemi, Slavica Jonic, Isabelle Rouiller

**Affiliations:** 1Department of Biochemistry & Pharmacology, Bio21 Molecular Science and Biotechnology Institute, The University of Melbourne, Melbourne, VIC 3010, Australiarvuillemot@gmail.com (R.V.); 2Australian Research Council Centre for Cryo-Electron Microscopy of Membrane Proteins, Parkville, VIC 3052, Australia; 3Ian Holmes Imaging Centre, Bio21 Molecular Science and Biotechnology Institute, The University of Melbourne, Melbourne, VIC 3010, Australia; 4IMPMC-UMR 7590 CNRS, Sorbonne Université, Muséum National d’Histoire Naturelle, 75005 Paris, France; slavica.jonic@upmc.fr

**Keywords:** AAA+ ATPase p97, continuous conformational variability, single-particle cryo-electron microscopy, MDSPACE, partially symmetrical protein complexes

## Abstract

Single-particle cryo-electron microscopy (cryo-EM) has been shown to be effective in defining the structure of macromolecules, including protein complexes. Complexes adopt different conformations and compositions to perform their biological functions. In cryo-EM, the protein complexes are observed in solution, enabling the recording of images of the protein in multiple conformations. Various methods exist for capturing the conformational variability through analysis of cryo-EM data. Here, we analyzed the conformational variability in the hexameric AAA + ATPase p97, a complex with a six-fold rotational symmetric core surrounded by six flexible N-domains. We compared the performance of discrete classification methods with our recently developed method, MDSPACE, which uses 3D-to-2D flexible fitting of an atomic structure to images based on molecular dynamics (MD) simulations. Our analysis detected a novel conformation adopted by approximately 2% of the particles in the dataset and determined that the N-domains of p97 sway by up to 60° around a central position. This study demonstrates the application of MDSPACE in analyzing the continuous conformational changes in partially symmetrical protein complexes, systems notoriously difficult to analyze due to the alignment errors caused by their partial symmetry.

## 1. Introduction

The function of molecular complexes, comprising proteins, DNA, and RNA, can be described according to a series of biochemical reactions, with each reaction being an equilibrium reaction associated with two or more globally stable three-dimensional (3D) structures, or conformations, of the complex. Characterizing the biochemical reactions and the associated changes in conformation is critical to understanding how protein complexes function. In recent years, single-particle cryo-EM has emerged as a routine method for determining the 3D structures of biological macromolecules at a high resolution [1,2,3,4,5,6,7]. Its significant impact on drug design and development is evidenced by its incorporation into the pipelines of pharmaceutical companies [8,9,10]. The common approach to determining the structure of a protein complex at a high resolution using single-particle cryo-EM is to collect a set of cryo-EM images of many copies of the purified complex under defined experimental conditions; classify the particle images into discrete groups based on their similarity, determined using correlation-based alignment; calculate 3D reconstructions (EM maps) of the classified particles; and build atomic models of the structures in the EM maps [11]. Popular software suites that implement discrete 3D classification methods are Relion [12] and cryoSPARC [13]. Images of particles that are not in a well-populated conformation are ignored, and information about the equilibrium of the reactions in the sample of the grid is only partially assessed. Moreover, even after classification, flexible regions of the complex may not be sufficiently resolved in the EM maps to allow for subsequent model building. This problem is worsened when the geometry of the protein complex promotes alignment errors. A notorious example of “disappearing” densities during 3D reconstruction is that of the AAA ATPase p97, for which nearly one-third of the entire protein complex, 6 times 30 kDa (180 kDa), remains unresolved [14] or poorly resolved [15,16,17,18,19] after discrete classifications.

The AAA ATPase p97 is one of the most abundant proteins in the cell cytoplasm and plays a role in nearly all cellular processes by unfolding and disassembling proteins from large complexes and targeting them for degradation [20,21,22]. More than 30 proteins called co-factors participate in the recruitment of p97 to its different pathways [23,24]. Mutations in the p97 gene have been associated with the development of multisystem proteinopathy (MSP) in humans, with the pathological accumulation of protein aggregates in the brain, muscle, and bone [25,26,27,28]. p97 has also been identified as a target for the treatment of cancer and neurological, viral, and parasitic diseases [29,30,31,32,33,34,35,36]. p97 assembles as a homohexamer with a quaternary structure in the shape of a double-stacked ring in the absence of a substrate [14,37]. Each p97 promoter contains a ~30 kDa N-terminal domain and two ~30 kDa AAA domains, D1 and D2. In the absence of a substrate, the D1 and D2 domains assemble into two superposed rings, the D1 ring (assembled from D1 domains) and the D2 ring (assembled from D2 domains). Despite its substantial molecular weight of 536 kDa, determining high-resolution structures of the entire p97 complex has been challenging. The resolution achieved using X-ray crystallography was limited to 4.7 Å for the full-length wild type protein [37,38] and 3.3 Å for the full-length protein bearing stabilizing mutations [39]. Single-particle cryo-EM achieved a higher resolution of 2.3 Å [15]. However, these globally high-resolution maps suffered from poor local resolution in some regions, especially in the N-domain regions, and interpretation of the high-resolution maps required the use of rigid-body docking for positioning these domains without possible refinement. A high resolution was achievable for individual protein domains, including the N-domain at 1.55 Å [40]. However, the overall flexibility of the N-domains has limited the resolution of the entire hexameric complex. NMR spectroscopy confirmed the flexibility of the N-domains and showed that its dynamic properties were modified by disease-associated mutations located at the interface between the N- and D1 domains [41]. Analysis of the R95G mutant using both single-particle cryo-EM and NMR spectroscopy in the presence of ADP demonstrated consistency between the results of the two methods. These findings highlighted that, in this particular mutant, the N-domains were in equilibrium between co-planar and upward conformations [16].

Each hexamer contains 12 nucleotide-binding sites, six between the D1 domains and six between the D2 domains, all capable of ATP hydrolysis [42]. The N-domains surround the D1 ring and have been described in two different conformations, one co-planar to the D1 ring and the other above the D1 ring, commonly referred to as the down- and up-conformation, respectively. The movement of the N-domains between the two conformations is a rotation by ~75° and a displacement of ~14 Å [15,43]. Mutations associated with MSP modify the dynamic properties of the N-domains, which, in turn, influence the recruitment of specific cofactors [44,45,46]. Thereby, the recruitment of p97 to different cellular pathways depends on the position of the N-domains, with, for example, the co-factors p47 and Ufd1-Npl4 preferably binding to the up-conformation and the co-factor UBXD1 to the down-conformation [20,41,47,48,49,50].

The structures of p97 and its yeast homolog bound to substrates show a large change in conformation in the D1 and D2 domains, with splitting of the rings into a washer-like conformation and a staggered arrangement of the domains along the unfolded substrate [17,18,51,52,53,54]. Co-factors recruit the substrate to p97, yet how they promote substrate engagement remains unknown at the mechanistic level, as they are mostly not resolved in the substrate-engaged conformations due to their intrinsic flexibility.

To disentangle the conformational heterogeneity of the particles in cryo-EM data, traditional image analysis methods use discrete classification. This approach groups particles into a predefined number of classes based on their conformational similarity. Methods such as multivariate statistical analysis [55,56] and 3D variance [57], the incremental K-means-like method of unsupervised 3D sorting [58], global (three-dimensional) variability [59], and, more recently the use of maximum likelihood algorithms, [12,13,60,61,62] have been employed. The focus of discrete classification methods is to solve a few average conformations (class averages) at high resolution, which occludes rare conformations and produces a partial picture of the full conformational variability in the particle induced by gradual conformational changes, with many intermediate conformational states (also known as continuous conformational variability). Thus, the need for new methods to decipher continuous conformational variability in cryo-EM data was recognized [63,64]. New methods, specifically developed for analyzing continuous conformational variability, are able to characterize one particle conformation per particle image [65,66,67,68,69,70], which generates the so-called conformational landscape, produced by mapping the conformations onto a low-dimensional (usually 2D or 3D) space. The first such methods were published in 2014 [65,66], supported by the increased quality of cryo-EM images collected using direct electron detector devices. The broader development of such methods started more recently [67,68,69,70,71], supported by recent computer power improvements. Recent deep generative models, such as CryoDRGN [67,72] and 3D Flex [68], allowed modelling non-linear motion manifolds and predicting the model of each particle in the form of 3D density maps. When prior structural information is available, in the form of an atomic model or an EM map, this information can be used to help extract other conformations present in the given dataset. The development of this type of continuous conformational variability approach was initiated with the method called HEMNMA [66,73]. HEMNMA uses normal modes [74,75] to elastically deform a reference structure (an atomic or EM map) and iteratively match the deformed model to particles in an elastic 3D-to-2D alignment procedure. The speed of HEMNMA was improved by combining it with a residual deep neural network (ResNet) (DeepHEMNMA [69]). However, the large amplitudes of normal modes, due to the large amplitudes of conformational changes, may induce distortions in the molecular structure. We showed that such distortions can be avoided by combining normal modes with molecular dynamics (MD) simulation using a method called NMMD [76], which was inspired by the methods for MD-based fitting of cryo-EM maps [77,78]. Recently, NMMD was embedded into an iterative conformational landscape refinement scheme, generating a novel method called MDSPACE [70]. While other methods estimate the conformations in the form of density maps, which are then analyzed to build atomic models, MDSPACE directly estimates one atomic-scale conformation per particle image. MDSPACE was implemented in the open-source ContinuousFlex plugin for Scipion [73,79].

In this study, we analyzed the conformational landscape of p97 using MDSPACE. The study was conducted with the R155P mutant of the AAA+ ATPase p97 in the presence of ATPγS, with these conditions chosen to induce the up-conformation of the N-domains. Analysis of the conformation of the mutant using discrete classification in Relion and applying C1 and C6 symmetry showed the presence of conformational heterogeneity in the N-domains. Relion’s discrete classification applied at the monomeric level failed to characterize this heterogeneity. Applying MDSPACE to the hexamers, we detected in the dataset the presence of an unexpected conformation adopted by less than 2% of the particles. This conformation is similar to that of the split-washer conformation of p97 previously observed with a substrate. Moreover, MDSPACE at the monomeric level showed that the N-domains moved by 60° around the average up-position of 14 Å above the D1 ring previously detected in standard 3D reconstruction workflows. The orientations of the N-domains in all observed conformations of the R155P mutant were notably different from the D1 coplanar conformation. Our study revealed a range of rotations for the N-domains, spanning from 45 to 105°, compared to the coplanar conformation. Furthermore, the data analysis with MDSPACE at the monomeric level showed a certain level of motion coordination of these domains.

## 2. Results

In the absence of a substrate, the N-domains of p97 have been reported in two different conformations, co-planar to or above the ring assembly of the D1 AAA domain of p97 [20]. The movement of N-domains is nucleotide-dependent. With ATPγS bound to both the D1 and D2 rings of p97, the N-domains of p97 were resolved in the up-conformation in one single-particle cryo-EM study [15] and in both the co-planar and up-conformations in another cryo-EM study [80]. Mutations located at the N-D1 interface and associated with the development of multisystem proteinopathy have been reported to stabilize the up-conformation [19,43,50]. However, all maps of p97 suffer from similar averaging artifacts; namely, the densities of the N-domains are poorly defined due to their intrinsic flexibility around the table core formed by the D1 and D2 domains. In this study, we aimed to characterize the dynamic properties of the N-domains in the up-conformation and compared the performance of discrete classification methods to that of MDSPACE. We chose to conduct this comparison of cryo-EM data on the R155P disease-causing mutant bound to ATPγS because of the dominance of the up-conformation (Appendix A). We observed that the densities corresponding to the N-domains in p97R155P-ATPγS have a fuzzier appearance than expected, contrasting with the clearer densities of the D1 and D2 domains (Appendix A).

### 2.1. The N-Domains of p97-R155P-ATPγS Are Partially Visible Using the Methods of Global Reconstruction and Discrete Classification

We first calculated the 3D cryo-EM maps of p97-R155P-ATPγS using the 274,640 images selected according to 2D classification (Appendix A) using the homogeneous refinement procedure imposing C6 symmetry or not imposing symmetry (C1). These C1 and C6 maps of p97R155P-ATPγS showed a well-defined central core, corresponding to the two rings formed by the D1 and D2 domains of p97, and poorly defined densities corresponding to the N-domains (Figure 1). The overall symmetry of the C1 map was 3.9 Å. The densities of the secondary structures of the D1 and D2 domains were well defined. However, the densities corresponding to the N-domains in this map were hardly visible at a threshold of 0.133 (Figure 1A). At an intermediate threshold of 0.0804, the densities of the two N-domains were somewhat defined, and the densities of three additional N-domains started to appear (Figure 1B). The densities corresponding to the remaining N-domain only started to be visible at a very low threshold (0.0143), a threshold at which noise also started to show (Figure 1C). Imposing C6 symmetry improved the global resolution to 3.5 Å but did not improve the definition of the densities of the N-domains (Figure 1D,E). These densities are notably absent at a threshold (0.135) that distinctly contours the secondary structures of the D1 and D2 domains. They become visible at lower thresholds (e.g., 0.0915, Figure 1D,E) but lack sufficient definition to fit secondary structures. At a very low threshold (0.0453), these densities roughly correspond to the general shape of the N-domains. Their positioning aligns closely with the location previously determined using rigid-body docking and cryo-EM of the wild-type p97 protein ([5], pdb id 5FTN). Upon fitting the 5FTN model to our C6 map, the correlation between one N-domain of the 5FTN model and the C6 map is 0.7865. This correlation improves to 0.8057 when adjusting the model’s N-domain to better match the density in the C6 map (with a 1 Å shift and 3° rotation).

The local resolution of the C1 map indicates higher resolution in the central densities corresponding to the D1 and D2 rings compared to the N-domains (Figure 1F,G). The low local resolution in the N-domain regions is coherent with their observed appearance/disappearance in the EM map and may suggest the strong flexibility of the N-domains.

### 2.2. Focused Classification Failed to Disentangle and Resolve the Different Positions of the N-Domains

We hypothesized that the reason for the lack of definition of the N-domains is caused by their intrinsic flexibility. We first performed 3D classification as implemented in Relion [81], focusing on the signal from a single monomer, with the particle images processed using signal expansion based on C6 symmetry and signal subtraction (Figure 2A). Three rounds of 3D classification were performed (Figure 2B). After the first round of classification, the densities of the N-domains were visible in only two classes (classes containing 317,564 and 309,759) and extremely poorly defined or completely absent in the other four classes. We grouped the particles belonging to the two classes with defined densities for the N-domain and repeated the process of classification with six classes. In the second round of classification, the N-domains were also visible in only two classes, this time containing 115,727 and 50,431 images, respectively (or 166,158 in total accounting for ~10.1% of all monomer images). Conducting a third classification after regrouping particles belonging to these two classes led to six classes, all containing the densities of the N-domains. Consequently, particles from the two classes of the second round of classification were utilized in a final refinement process. This procedure enabled us to improve the resolution of the monomeric map from 3.9 Å to 3.0 Å (Figure 2C,D).

We also grouped the particles belonging to the four classes with no or poor densities for the N-domain and submitted these to another round of classification and repeated the classification process. However, despite several rounds of classification, we were unable to retrieve the densities of the N-domains by regrouping the particles from the classes in which this domain was not visible. The inability to resolve the densities of the N-domains suggests that the movement of the N-domains may not be discrete and could be continuous.

### 2.3. Focused Classification on Low-Pass Filtered Images Detected Movement of the N-Domain by Up to 30 Degrees

We reasoned that the challenge in resolving the N-domains during focused classification might be attributed to substantial large-scale motions. To explore the presence of such motion, we conducted an analysis on low-pass filtered images using the focused classification. Low-pass image filtering attenuates high frequencies, such as noise and the structural details of the particle, while retaining the low-frequencies associated with the global shape of the particle and any significant global motions it may exhibit.

We pre-processed all 274,640 particles (Appendix A). All the particles were low-pass filtered inside Scipion using the Xmipp protocol [82] at a cut-off frequency of 6 Å. At this resolution, alpha helices can still be distinguished [83]. Then, the maximum likelihood classification approach in Relion was performed, using 10 classes (step 1, Appendix A) [38]. Particles belonging to class number 5 (158,000 particles) were selected for further analysis, as this class contained the majority of the images and showed reasonably well-defined N-domains. A separate Relion auto-refinement of this map using 158k unfiltered particles and without imposing symmetry resulted in a map with the six N-domains visible in the up-conformation (step 2, Appendix A).

After symmetry expansion and signal subtraction, a monomer-focused classification was performed with seven classes (step 3, Appendix A). The classes displaying the densities of the N-domain exhibited variation in the positioning of this domain. The most pronounced divergence in the positioning of the N-domains was observed between classes 6 and 7 (Appendix A), revealing an approximate 30-degree tilt (Figure 3A). This angle measurement was taken from the point where the N- and D1 domains connected to the center of mass within the N-domain density. Another class (class 2) showed the densities of the N-domain in the same position as the 5FTN model, with densities corresponding to the N-D1 linker and the close proximity of the N- and D1 domains (Figure 3B).

Three classes (classes 1, 4, and 5, Appendix A, step 3) displayed no discernable densities of the N-domains. Upon regrouping these three classes and conducting another monomer-focused classification, asking for the monomer classification in three classes (step 4, Appendix A), two of these classes exhibited the densities of the N-domains. However, the third class, comprising approximately 206,000 particles, still did not display visible N-domains, indicating a higher degree of conformational flexibility. Of note, none of the classes obtained in this analysis detected the N-domains in the co-planar position with the D1 domain.

Together, the focused classification on low-pass filtered images successfully segregated hexamers and monomers displaying different conformations, highlighting significant large-scale conformational variability within the N-domains. However, this approach did not comprehensively capture all the conformations adopted by these domains. The monomer-focused classification on its own was unable to retrieve useful information from the particle images for inferring distinct and visible N-terminal domain conformations.

While an analysis involving a substantially larger dataset combined with extensive classification might offer more insights into the diverse positions and motions of the N-domains within the protein complex, this approach may still not fully sample their entire range of motion.

### 2.4. Continuous Conformational Variability Analysis Using MDSPACE at the Hexameric Level

We next explored whether MDSPACE, our recently developed image analysis method [70], could effectively evaluate the entire conformational spectrum of p97R155P-ATPγS, despite the apparent high (C6)-symmetry core of the complex made by the D1 and D2 rings. The resulting set of atomic coordinates, fitted to the particle images using MDSPACE, was analyzed using UMAP to generate a conformational landscape spanning ten dimensions. The two first dimensions are shown in Figure 4A.

The resulting two-dimensional scape revealed two distinct clusters: a predominant large cluster comprising 98% of the particles and a smaller cluster housing 2% of the particles. These two clusters were identified manually based on the density of the points in the two-dimensional space. The average atomic coordinates from these two first clusters (cluster-averaged structures) are presented in Figure 4B,C, respectively. These structures are shown in color based on the values of the root mean square fluctuation (RMSF), which measures the average variation in each residue with respect to the cluster-averaged structure.

The RMSF of the D2 and D1 rings for the large cluster is low (below 2 Å), indicating the strong stability of the D1 and D2 rings, whereas the RMSF of the N-domains is higher than 10 Å, indicating high conformational variability. The small cluster shows a similar variability in the N-domains (an RMSF above 10 Å) but less stable D1 and D2 rings (RMSF ranges from 3.5 Å to 7.5 Å).

The averaged structure of the small cluster shows a striking opening between the D1 and D2 rings (Figure 4C). This particular conformation has not previously been identified in datasets of p97 or its yeast homolog cdc48. However, it bears resemblance to an open conformation obtained in the presence of a substrate (EMD-23450, pdb 7LN6 [54]). The small cluster comprises 5475 images, roughly 2% of the dataset. Due to the limited number of images and the flexibility observed within this class, the resolution of the map generated with these particles is insufficient to discern the presence or absence of an unfolded polypeptide chain at the center of the spilt hexamer, as was observed and modeled in EMD-23450 (pdb 7LN6).

The first two UMAP components allow us to separate global changes such as the opening of D1 and D2 but are not sufficient to fully describe the variability in the N-domains present in the structures placed in the large cluster. Therefore, we analyzed the ten computed UMAP dimensions and identified several local minima in this ten-dimensional space. Six examples of such sub-clusters are shown in Figure 4A,D. These sub-clusters are low-energy regions in the ten-dimensional space. They were selected manually and arbitrarily, each containing approximately 10,000 particles. These six sub-clusters were projected onto the two-dimensional space and encircled in Figure 4A. These sub-clusters reveal the diversity of the mobility of each of the six N-domains. The displacement of the N-domain around its position in the 5FTN model (considered to be the zero-degree position) by up to +/−30° in each of the six sub-clusters is schematically represented in Figure 4D. Of note, the positions of the six N-domains are not identical within a single hexamer.

Still, the 3D reconstructions calculated from the particles in the sub-clusters contained partly visible N-domains, indicating that some heterogeneity remained in the sub-clusters. Indeed, if we consider each of the six N-domains moving up and down gradually and independently of the other N-domains, this will result in complex variations and a very high number of conformational states. Although this analysis using the ten-dimensional UMAP space allowed us to well separate conformational states with large-scale differences, such as those related to the motions of the D1 and D2 rings and some of the main motions of the N-domains, it did not allow us to obtain fully homogeneous sub-clusters, with this high-order pseudo-symmetric (C6) complex adopting independent small-scale motions in multiple (six) regions. Further clustering in a high-dimensional UMAP space might better separate the different conformational states. However, such an analysis would result in an excessively large number of clusters and would be impractical. A sufficient number of particle images should be available per cluster in order to obtain 3D reconstructions of sufficient quality to validate the analysis. Another consideration is that the UMAP embedding does not account for rotations around the C6 pseudo-symmetry axis. Indeed, one particular conformational state could be repeated six times in the data (*n* × 60° rotations around the C6 symmetry axis, *n* = 6), which increases even more the complexity of the embedding.

Here, it should be noted that, before using the UMAP, the Cα models fitted to the particle images were rigid-body-aligned with respect to the C6-symmetrical model, with the N-domains in the up position (Cα model obtained from PDB 5FTN). This conformation was used to start the MD simulations in MDSPACE. Thus, the rigid-body alignment of the fitted asymmetrical models with the C6-symmetrical model may not result in perfectly aligned copies of the states that only differ in the position of the N-domain in the ring, which induces additional difficulties in separating the truly different states using UMAP.

### 2.5. Analysis of the Continuous Conformational Variability at the Monomeric Level Using MDSPACE

Although MDSPCE analysis of the conformational variability of the entire p97 complexes allowed us to detect the N-domain motion complexity at the hexameric level, it did not allow us to accurately characterize the motion of each monomer. To obtain a more detailed description of the N-domain conformations, each hexameric structure was split into six monomeric structures, which were aligned and embedded into a new conformational landscape using UMAP, shown in Figure 5A. The conformational landscape of the monomers reveals a single dominant motion (half-circle trajectory in Figure 5A) with a central low-energy region (high density of particles). By selecting regions along this trajectory (shown with the black dotted line boxes in Figure 5A), we identified several positions of the N-domains. The structures associated with the boxed regions in Figure 5A were averaged to produce the atomic structures presented in Figure 5B. The central region (Figure 5A) matches the initial conformation of p97 with the N-domain up (PDB 5FTN) and was therefore annotated as “0°” (tilt). The left and right boxes along the trajectory are associated with the downward and upward rotation of the N-domain, respectively, with a maximum of approximately 30° on each side. The averaged atomic structures (Figure 5B) clearly show that even though the N-domains are flexible, they do not adopt the D1 co-planar conformation (PDB 5FTK or 5FTM), which could correspond to a rotation of about −75°. This analysis shows that the N-domains of the mutant are consistently above the D1 ring, exhibiting a rotation ranging from 45 to 105° compared to the coplanar conformation.

The particles corresponding to the dotted boxes in Figure 5A were used to reconstruct the EM maps shown in Figure 5C, where the densities of the entire range of motion of the N-domain were now well defined, revealing a range of rotation of about 60°.

### 2.6. Retrieving Hexameric Conformations from the Analysis at the Monomeric Level

The analysis of the conformational landscape presented in the previous section allowed us to accurately define the conformational state of each monomer in each particle image. To evaluate whether the movements in the N-domains exhibited coordination within the hexameric complex, we reported the individual conformational state based on its corresponding location within the hexameric structures. This analysis aimed to ascertain any potential coordination or synchronicity in the movements observed among the N- domains within the context of the hexameric assembly.

Considering the continuous nature of the obtained conformational landscape (Figure 5A), it is not possible to obtain clearly separated clusters (discrete classes). Therefore, we decided to discretize this conformational landscape into an arbitrarily chosen number of regions (three) along the first UMAP component, with each region containing the same number of particles. Because the first UMAP component corresponds to the principal axis of motion of the monomers, dividing the landscape into three regions with equal number of particles results in regions containing structures with the rotation of the N-domains ranging from −30° to −10°, −10° to +10°, and +10° to +30° relative to their position in the p97 up-conformation (PDB 5FTN). We annotated these three regions by −20°, +0°, and +20°, respectively (Figure 6A), representing the approximate average rotation angles of the N-domains within the structures in these regions. Here, it should be noted that 0° corresponds to the position of the monomeric N-domain in the p97 up-conformation (PDB 5FTN). Subsequently, each monomeric state (determined with the labels −20°, 0°, or +20°) was mapped onto its location within the hexamer structure. This process resulted in 130 unique combinations of N-domain conformations across the entire dataset of 242,130 particles (Figure 6B). This analysis allowed us to visualize the distribution of these various conformational states within the hexameric assembly (Figure 6C). This histogram gives a description of the distribution of the conformational states present in the data at the hexameric level, revealing some favored relationships between the N-domains compared to others. The most present conformational states (with more than 4000 particles per state) show five N-domains in the same position and one N-domain in another position or four adjacent N-domains in one position and two in another position. On the contrary, the least probable conformations are the states where each adjacent N-domain is in a different position, with less than 10 particles per state.

## 3. Discussion

Thorough characterization of the structural flexibility of biomolecules is critical to understanding their functional molecular mechanisms. Single-particle cryo-EM presents an unparalleled avenue for exploring this flexibility since purified molecules prepared for cryo-EM are not confined to specific conformations, providing an in-depth insight into their structural dynamics. In this article, we presented a study of the continuous conformational changes adopted in a partially symmetrical protein complex, a notably challenging task due to the interpretation errors stemming from its partial symmetry. Our test sample was the AAA ATPase p97 bearing a disease-associated mutation (p97-R155P) bound to ATPγS, a non-hydrolysable ATP analog, as this complex favors a set of particular conformations, with the N-domains above the D1 ring, while adopting challenging C6 partial symmetry. Despite the tendency of the N-domains of this mutant to adopt a preferential upward conformation, the densities corresponding to the N-domains were poorly defined (Figure 1), suggesting a broad spectrum of N-domain conformations. The primary goal of this study was to characterize the N-domain flexibility and evaluate the performance of the recently developed MDSPACE method on this challenging problem, considering that MDSPACE was specifically tailored to addressing the continuous conformational heterogeneity of biomolecular complexes in cryo-EM datasets.

Initially, we analyzed the conformational variability in p97-R155P-ATPγS using focused 3D classification, concentrating on the N-domain region, in conjunction with symmetry expansion and signal subtraction. Following an extensive classification process (Figure 2), we obtained two distinct types of maps of the p97-R155P-ATPγS monomer: one exhibiting visible N-domains and the other lacking density. Through the regrouping of monomers displaying similar N-domain conformations, the resolution of the map improved from 3.9 Å to 3.0 Å. However, this approach did not enable the segregation of significantly different conformations from the average conformation: these particles were grouped together during the discrete classification, resulting in the averaging-out of the N-domain densities. While focused 3D classification is a method that has demonstrated efficiency in cases involving compositional heterogeneity [84] and distinct clustered conformations [16,41,85], it showed severe limitations when confronted with continuous heterogeneity, as observed in this study. The use of focused classification with low-pass filtered images showed slight improvement. It facilitated the identification of large-scale movements (Figure 3) but still fell short in fully characterizing the entire range of motion.

The image analysis conducted using MDPSACE enabled the identification and characterization of a continuum of conformational states within the N-domain of p97. Our examination of the conformational space at the hexameric level revealed that 98% of the data exclusively encompassed variations in the N-domains (Figure 4). Although the analysis at the hexameric level identified variability within the N-domain, it could not fully capture the intricate complexity of these variations accurately. The analysis of the variations at the monomer level using MDSPACE showcased the continuous spectrum of conformations, revealing a swing of 60° in the N-domains around a central position (Figure 5). This central position closely resembles the conformation resolved at a higher resolution during focus classification (Figure 2C, pdb id: 5FTN). This conformation was the conformation determined using focused classification and was used as the reference for starting the MD simulations in MDSPACE. Furthermore, our analysis revealed that the most probable conformational states occurred when the neighboring N-domains were in similar positions, irrespective of whether they were oriented around the 0° position, closer to −30°, or closer to 30°. Approximately 27% of the hexamers had five N-domains in similar positions. This distribution is coherent with the global map (Figure 1), which was calculated with all the particles (Figure 1C). The least probable states were those with neighboring N-domains in different conformations. Although an explanation for this phenomenon is currently lacking, it is noteworthy that several factors involved in recruiting substrates to p97, such as p47 [86] and UBXD1 [85,87], interact with adjacent N-domains and that, in the recently published structure of p97 involved in the ERAD pathway, the four p97 N-domains interacting with Derlin-1 were in a different position [88].

The MDSPACE method also detected a conformation adopted by 2% of the particles in our dataset, wherein the two rings of p97, the D1 and D2 rings, displayed an asymmetrically open conformation (Figure 4C). This conformation resembles a previously detected conformation observed in the presence of a substrate [54]. Comprehending the significance of this conformation necessitates additional data. This could potentially arise due to a small percentage of p97 unfolding or complexing with contaminants introduced during purification. Conversely, it might represent the dynamic state of p97 in the process of monomer exchange and/or ring opening, in a conformation potentially ready to engage and be stabilized by cofactors. Further analysis is required to assess the significance of this conformation. This includes investigating a much larger dataset to calculate the 3D EM map of this conformation at a resolution sufficient to detect the presence or absence of a polypeptide chain in the central pore of p97. Additionally, exploring various mutants of wild-type proteins in the presence and absence of co-factors is required for a comprehensive understanding. Validation through single-molecule fluorescence studies would also help assess the dynamic properties and tendency of p97 to adopt this D1-D2 ring open conformation. It is possible that the proportion of p97 in the asymmetrically open D1-D2 ring conformation is higher for the R155P mutant compared to the wild type, considering the significantly heightened ATPase activity observed in this mutant [89] and its differential interaction with co-factors [47]. Nonetheless, this outcome underscores the MDSPACE method’s efficacy in revealing distinct conformational information that diverges significantly from the initial model. This discovery presents captivating avenues for further exploration and investigation.

## 4. Materials and Methods

### 4.1. Sample Preparation

The construct pTrcHisB-p97R155P with an N-terminal His tag used in this study has been previously described [89]. The proteins were expressed in BL21 (DE3) Escherichia coli cells and purified using nickel affinity (HisTrap, GE Healthcare, Chicago, IL, USA) and size exclusion chromatography with a pre-calibrated 24 mL Superose 6 increase 10/300 gel filtration column (GE Healthcare Biosciences). The protein concentration was measured using the NanoDrop (Thermo Fisher Scientific). A total of 3 mg/mL of the purified protein, p97WT and p97R155P, was incubated with 5mM ATPγS (Sigma, Sofia, Bulgaria) for 30 min at 4 °C. To promote multiple orientations of p97 in the EM grid, 0.02% NP40 was added immediately before freezing, and then 4 µL of the protein was spotted onto the negatively glow discharged EM holey carbon grids (Quantifoil R1.2/1.3, 200 mesh, Electron Microscopy Sciences, Inc., Hatfield, PA, USA), blotted for 3.5 s with a blot force = −1, and frozen-hydrated by plunging it into a bath of liquid ethane slush using the Vitrobot IV system (Thermo Fisher Scientific) at 4 °C and 100% humidity.

High-resolution images of p97R155P-ATPγS were recorded on the Talos Arctica 200 kV (Thermo Fisher Scientific Inc, Waltham, MA USA) equipped with a K2 direct electron detector (Gatan Inc, Pleasanton, CA, USA). EPU software (Thermo Fisher Scientific, Inc, Waltham, MA USA) was used for the data collection. The data were collected with a pixel size of 1.31 Å/pixel, at 100 k magnification, and with a cumulative exposure of 50 e^−^/Å^2^. A total number of 2359 movies were collected with 40 frames per movie.

### 4.2. 3D Reconstruction

Image processing was performed using cryoSPARC v2 [13]. Movies were motion-corrected using patch motion correction. Then, the motion-corrected micrographs were used for contrast transfer function estimation with patch CTF estimation. An automated particle selection was made using a template picker and 1,103,098 particles picked and extracted using a 200-pixel box size. These particles were used for 2D classification (Appendix A). After 2D classification, 274,640 particles were selected for homogenous refinement. The resolution of the map obtained without applying symmetry (C1) was estimated to be 3.9 Å using the gold-standard Fourier shell correlation, and that of the map calculated applying C6 symmetry was estimated to be 3.5 Å (Figure 1). Heterogenous Refinement/3D variability analysis in cryoSPARC v2 [71] and 3D discrete classification in Relion v3 [60] were attempted. However, the resulting maps exhibited similarities to those obtained through homogeneous refinement. Notably, the C6 maps displayed poorly resolved densities corresponding to the N-domains, and the C1 maps showed the unequal distribution of the densities of the N-domain and a similar resolution. Repeated classifications failed to generated classes with the N-domains well defined and in different positions.

### 4.3. Focused 3D Classification at the Monomeric Level

To isolate the signal corresponding to each monomer within the hexameric particles, a multi-step process was employed using the procedure previously described [60]. The 274,640 particles from the 3D refinement were artificially expanded six times by performing 60° rotations according to the pseudo-symmetric point group (C6) to obtain 1,647,840 particles. A projection of p97 in the orientation determined during alignment, in which a monomer was masked, was subtracted from the particle image. The initial mask used for the particle subtraction routine was created by combining the PDB files of the p97 monomer with the N-domain in the up- and down-conformations (pdb ids 5FTN and 5FTK, respectively [15]). This combined PDB file was transformed into an EM map in ChimeraX v1.2 [90], and a mask was created by applying a 15 Å low-pass filter with a two-pixel soft edge (Figure 2A and Appendix A). The second mask encompassed the entire particle with its N-domain either in the up- or down-conformation, excluding the region corresponding to one monomer. This second 3D mask was applied to the C6 map obtained from cryoSPARC (Figure 1D,E) using Xmipp in Scipion [79,91,92]. Projections of this masked reconstruction were subtracted from all the symmetry-expanded particle images. This procedure effectively removed the signal for five of the six monomers from the C6 map, resulting in particle images containing a signal for only one monomer.

Focused 3D classification was performed using Relion on these expanded–subtracted sets of particles without performing further alignment and using an initial reference map low-pass filtered to 60 Å. The particles were first classified into six classes. Then, the particles from similar classes were merged and subjected to another round of classification. This step was repeated for each round of 3D classification until no further improvement was observed in classification (Figure 2). Finally, auto-refinement was performed using the particles included in each class, excluding shift search and limiting the angular search to 1.8 degrees.

The same focused classification approach was also performed with expanded–subtracted images that had been low-pass filtered with a cut-off value of 6 Å using the same approach described above.

### 4.4. Analysis of Conformational Heterogeneity Using MDSPACE

MDSPACE [70] was used to investigate the continuous conformational variability in the N-domains of p97. MDSPACE was applied to the particles that were pre-aligned using the rigid-body alignment parameters determined during the C1 CryoSPARC homogenous refinement. This initial alignment was refined using MDSPACE. Images with C1 alignment parameters were binned by 2 to a pixel size of 2.62 Å^2^. Two iterations were performed, iteration 1 with 5000 particles randomly selected and iteration 2 with the entire dataset of 272,640 particles. The structure of the p97 WT with the N-domains in the up-conformation was used (pdb id: 5FTK, [15]) as the initial conformation for the NMMD fitting [76] within MDSPACE. The normal modes used in NMMD were computed from the initial conformational state, and a subset of the five lowest frequency normal modes was selected for the NMMD fitting. The five lowest frequency normal modes were included for fitting. NMMD includes MD simulations, which were performed using a Gō-like coarse-grained model (considering only Cα atoms) for 50 picoseconds, with a time step of 1 femtosecond, a temperature of 50 K, and a force constant of 4000 kcal/mol. A coarse-grained model was selected to speed up the MD simulations, considering the large size of the p97 and one MD simulation per particle image run using MDSPACE. This approach significantly reduced the computational cost while ensuring accurate simulation of the native dynamics.

The entire MDSPACE workflow was executed using the ContinuousFlex plugin for Scipion [93]. The analysis of 32,510 particles failed due to particles that either were incorrectly picked or contained an insufficient signal for the fitting to converge. These failed fitted structures and images were removed from the analysis. The remaining fitted structures (Cα atomic coordinates) were then rigid-body-aligned to discard the rigid-body motions induced during the MD simulations, as described in the original MDSPACE publication [70]. The rigid-body alignment was performed with respect to the Cα atomic model that was used as the initial conformation for the NMMD fitting in MDSPACE. Uniform Manifold Approximation and Projection (UMAP) [94] was then applied to the Cartesian coordinates of the Cα atoms of the set of fitted and rigid-body aligned structures obtained using MDSPACE to project them onto a low-dimensional conformational space. Compared to other approaches, such as principal component analysis (PCA) [95], which decomposes the variability into a linear combination of principal components, UMAP is a dimension reduction method that captures non-linear features in the data, which in practice may allow a better separation of the different conformational populations. The clustering in the UMAP-based low-dimensional conformational landscape of the hexameric p97 was performed manually by considering the density of points in different regions of the landscape (interactive grouping of points in dense regions). Due to the complexity of the conformational variability that was detected during this analysis, the conformational variability was further analyzed at the monomeric level by extracting the co-ordinates corresponding to each monomer from the set of the finally fitted structures, resulting in a set of 1,440,780 monomer structures. This set of monomeric structures was rigid-body-aligned with a reference monomer by aligning the D1 and D2 domains only (excluding the N-domains). The UMAP dimension reduction was applied to the rigid-body-aligned monomer structures, and manual clustering was performed in the obtained UMAP-based low-dimensional conformational landscape first (through interactive grouping of points in dense regions). Then, the obtained conformations of the monomers were mapped back onto their positions in the hexamer. To this end, the monomeric conformational space was first discretized along the first UMAP component (which corresponds to the principal axis of motion) to obtain three regions with the same number of particles. Finally, the conformation of each monomer in these three regions was mapped onto the position of the corresponding monomer in the hexameric p97.

In this article, the conformational landscape is shown in terms of free energy difference. The density of the points in the low-dimensional space obtained using UMAP is converted into free energy differences using the Boltzmann factor $\Delta G/k_{B}T=-ln\left(n/n_{0}\right)$ by counting the number of particles n in each region of the space and the number of particles in the most populated region $n_{0}$, with ${\;k}_{B}$ being the Boltzmann constant and T the temperature of the system.

## 5. Conclusions

In conclusion, while methods like focused classification detect the presence of conformational heterogeneity, their ability to capture the full spectrum of conformations is limited. In contrast, MDSPACE has emerged as a robust approach, offering extensive and detailed insight into the diversity of conformations adopted by a protein complex. Even within the complexities of a partially symmetrical structure, MDSPACE has demonstrated its capability to unveil a myriad of conformations. MDSPACE also identified a distinct conformation within a very small subset of particles (less than 2%), highlighting the proficiency of this method in discerning possibly rare structural states, even when they differ significantly from the initial model.

## Figures and Tables

**Figure 1 ijms-25-03371-f001:**
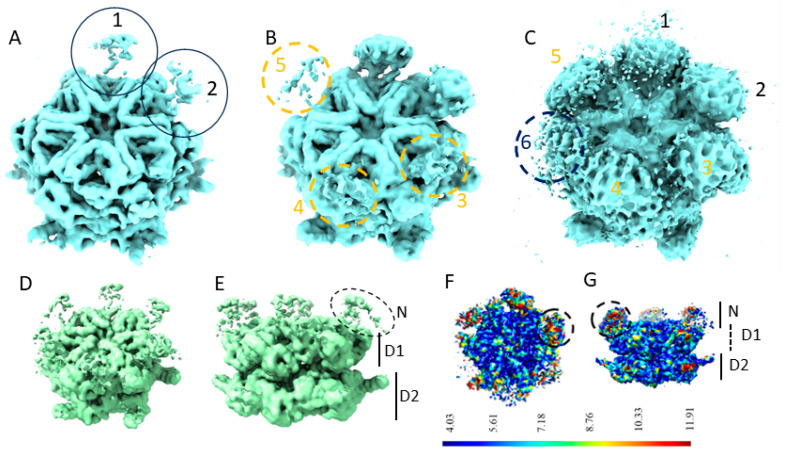
Poorly defined N-domains of p97R155P-ATPγS when calculated using homogeneous refinement (**A**–**C**). A 3.9 Å EM map of p97R155P-ATPγS calculated without imposing symmetry shown at thresholds of 0.133 (**A**), 0.0804 (**B**), and 0.0143 (**C**). The positions of the six N-domains are indicated and numbered in the order in which they become visible when decreasing the threshold. (**D**,**E**) A 3.5 Å EM map of p97R155P-ATPγS calculated with imposing C6 symmetry, with the position of the N-domains and the rings formed by the D1 and the D2 domains indicated, shown at a threshold of 0.0915. (**F**,**G**) Local resolution of the C1 map.

**Figure 2 ijms-25-03371-f002:**
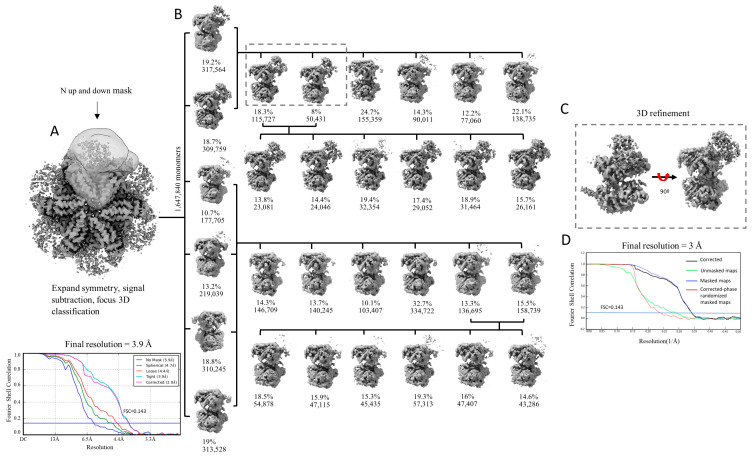
Focused 3D classification of p97R155P-ATPγS. (**A**) Three-dimensional reconstruction calculated imposing C6 symmetry and mask used for particle signal subtraction and classification with FSC curves. (**B**) Focused 3D classification of the selected particles after symmetry expansion and signal subtraction. Three rounds of classification were performed (round 1: vertical axis; round 2: first row; round 3: second row). For rounds 2 and 3, classification was performed with the particles grouped from the previous classification according to the presence or absence of density of the N-domains. (**C**) Three-dimensional refinement of the classes with homogenous particles, which led to the final monomer map with 3 Å resolution. (**D**) FSC plot.

**Figure 3 ijms-25-03371-f003:**
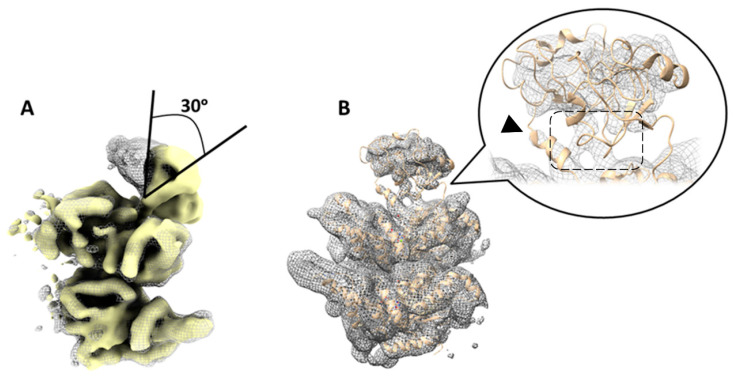
Variability in the position of N-domains of p97R155P-ATPγS detected using focused classification on low-pass filtered images. (**A**) Superposition of two different class averages (yellow solid and gray mesh) corresponding to classes 6 and 7 in Appendix A. This comparison highlights a noticeable tilt in the N-domain by ~30 degrees between these two classes. (**B**) Another class (class 2 in Appendix A, step 3, gray mesh) displaying the N-domain in a similar position to the published 5FTN model (beige cartoon). Densities of the N-D1 linker are resolved (arrowhead). The N- and D1 domains are in close proximity to each other (dashed box).

**Figure 4 ijms-25-03371-f004:**
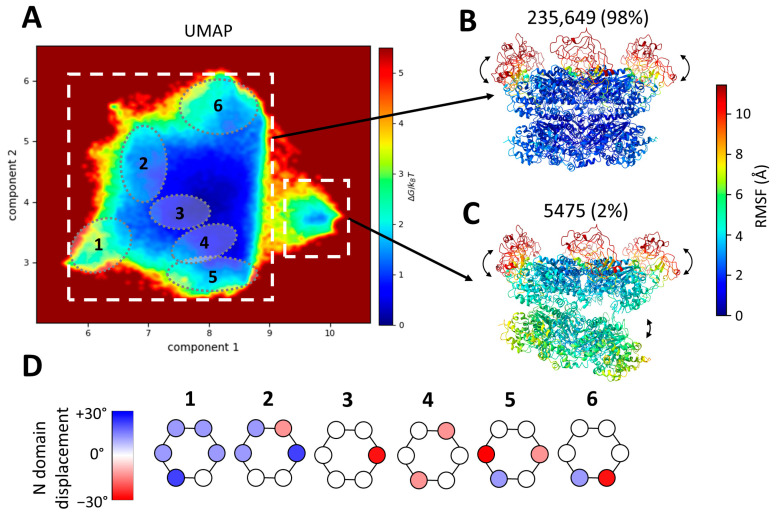
Conformational landscape of p97R155P-ATPγS obtained using MDSPACE at the hexameric level and example structures identified in this landscape. (**A**) Conformational space obtained using MDSPACE and UMAP. Two clusters can be distinctly separated (two white squares) and are presented in (**B**,**C**). Six sub-clusters showing examples of N-domain variation are encircled and presented in (**D**). (**B**,**C**) Average atomic structures of the two clusters designated by white squares in (**A**). The colormap shows the average variation in each residue (RMSF) with respect to the cluster-averaged structure. The first cluster (**B**) shows the strong stability of the D1 and D2 rings combined with the flexibility of the N-domains. The second cluster (**C**) shows the flexibility of the N-domains combined with a gap opening between the D1 and D2 rings. (**D**) Displacement of the N-domain around its position in the 5FTN model (considered to be the zero-degree position) by up to +/−30^°^ in each of the six sub-clusters shown in (**A**).

**Figure 5 ijms-25-03371-f005:**
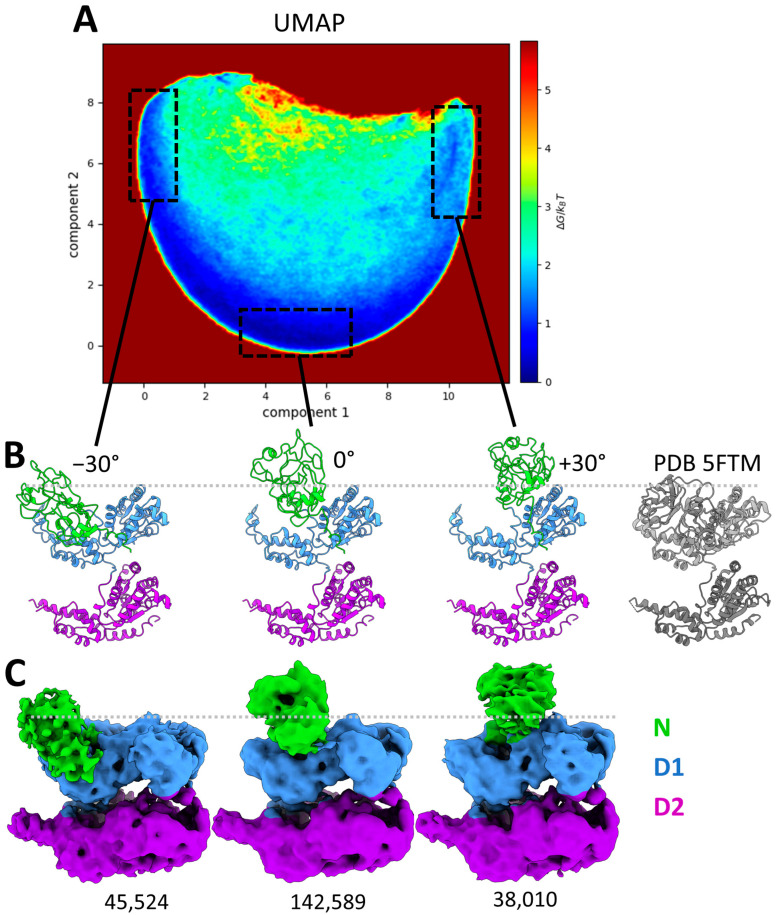
Conformational landscape of p97R155P-ATPγS obtained using MDSPACE at the monomeric level. (**A**) Conformational landscape shown using the two first components obtained using UMAP, in the shape of a half-circle trajectory along the first component axis. Three regions, boxed in black dashed lines, correspond to the edges and the middle of the trajectory. (**B**) The averaged atomic structures of three regions boxed in (**A**) showing angular displacement of the N-domain of −30°, 0°, and +30°, respectively, and the PDB 5FTM structure of p97 in coplanar conformation (gray). (**C**) Three-dimensional reconstructions calculated from the images in three boxed regions shown in (**A**), corresponding to the three averaged atomic structures shown in (**B**). The number of particles used for the reconstruction is written below each reconstructed EM map.

**Figure 6 ijms-25-03371-f006:**
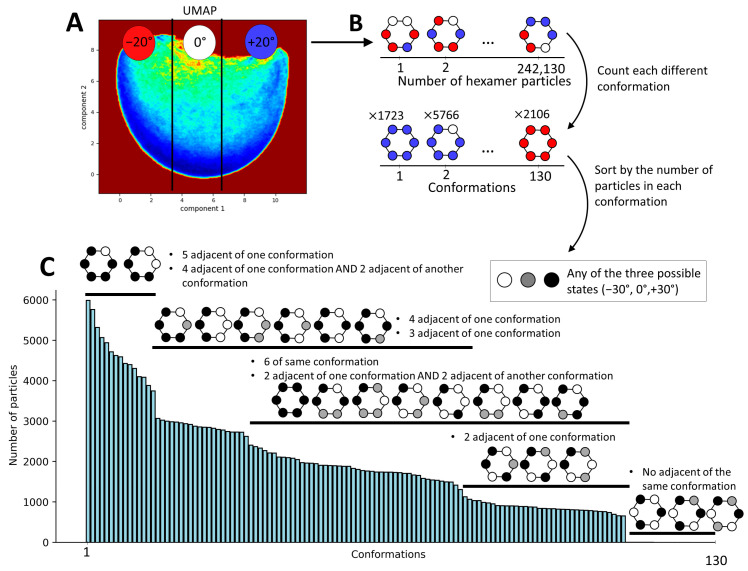
Determination of the hexameric conformation index of p97-R155P-ATPγS based on the monomeric conformational states obtained using MDSPACE. (**A**) Conformational space at the monomeric level obtained using UMAP, split into three regions along the first UMAP component so as to obtain an equal number of particles in each region. These regions are labeled as −20°, 0°, or +20°, representing the approximate average positions of the N-domains with respect to the up-conformation. (**B**) Schematic of the process undertaken for reporting the conformation of each monomer according to its corresponding hexamer structure. (**C**) Histogram of all hexameric conformational states derived from the monomeric conformational states observed in (**A**). The number of particles for each distinct state is showed in the histogram. The black and white schematic identifies similarities among these states, highlighting commonalities regardless of the individual group affiliations of the particles.

## Data Availability

The software code for MDSPACE is available on GitHub (https://github.com/scipion-em/scipion-em-continuousflex), deposited 16 October 2023 and is also part of the open-source ContinuousFlex plugin of Scipion v.3.

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
