# Peer review of "Analysis of the Conformational Landscape of the N-Domains of the AAA ATPase p97: Disentangling the Continuous Conformational Variability in Partially Symmetrical Complexes"

_ijms, 2024, doi:10.3390/ijms25063371_

Round 1

Reviewer 1 Report

Comments and Suggestions for Authors

The work from Valimehr et al. deals with the use of Cryo-EM to study the dynamic hexamer of a p97 mutant known to stabilize a upward conformation of the N-terminal domain. Despite such stabilization, the authors found the N-terminal domain of each protomer is still highly flexibile and the use of the focused classification is not sufficient to catch and fit different clusters for the hexamer and monomer with different combinations/conformations for the N-terminal domain orientation. This and the low resolution of many particles from EM maps suggested that there is a continuum of conformations. Therefore they relied on MDPSACE to show how it improved the identification and characterization of this a continuum of conformational distribution at the hexamer level, which allowed the definition and modelling of different N-terminal orientations for monomers; these different orientational monomers allowed to ascertain a preferential orientation of N-terminal domain of adjacent subunits in the hexamer. Moreover MDSPACE, by allowing a superior signal-to-noise ratio when comparing MD-fitted 3D images, found a rare conformation of the hexamer that is open in the middle.

The work is technically well performed and decently presented. Apart from the full list of points found below, there is a major point to adress: cosidering the continuum distribution of conformations that was obtained in the free energy landscape of the monomers reconstructed using MDSPACE, it is not possible to actually define discrete conformations (as it is evident in Fig. 6), so the authors provided an arbitrary classification to estimate the statistical preference of certain conformations in the hexamer. This is not hampering the conclusion of the work, but this legitimate arbitrary criterion (as there is a continuus, it is possible to make a classification with arbitrary geometric criteria over this continuum) has to be declared, or the authors has to demonstrate that it is the result of an automatic clustering algorithm (to be added in the methods section) that operates in a multi-dimensional space.

List of points:

- Line 107: "...iteratively match the deformed model to particles..." The particles that are intended are the 2D Cryo-EM images?

- Line 129: the rotation was previously described as known [5,9]. What is the difference detected by the authors with MDSPACE? It seems to be the big rotation for the N-terminal stabilized mutant; anyway, this aspect is not underlined or valorized in the manuscript

- Line 231: Which is the input vector to UMAP? RMSD distance between all atoms (which atoms? alpha Carbon?), torsion angles? Inter-residue distances?
UMAP is not a clustering technique. What is the clustering technique used?

- Lines 260-262:"these densities also had a fuzzier appearance when compared to the N-domains in the projections of the PDB structures with the up and down N-domain conformations (Figure S1B,C)."
Not clear

- Lines 289-292:"The local resolution of the C1 map indicates higher resolution in the central densities corresponding to the D1 and D2 rings compared to the N-domains (Figure 1F–G). The low local resolution in the N-domain regions is coherent with their observed appearance/disappearance in the EM map and may suggest a strong flexibility of the N-domains." This is redundant

- Paragraph 3.2 can be simplified, as repeats many methodological steps previously reported

- Line 310: "Regrouping particles from the classes where the N-domain was resolved enabled us to improve the resolution of the monomeric map from 3.9 Å to 3.0 Å " In this group the N-terminal domain is visible on monomers? How many resampled/subtracted images belong to the first two classes in which the N-terminal domain is solved? This is important to understand the degree of uncertainity in the classification and how much continous is the conformational shift

- Line 328: why only "class 5" molecules? Are they better resolved? Because it was known they are up-conformation?

- Lines 358-359:"However, this approach did not comprehensively capture all potential conformations of these domains." I would say the main point is that using a monomer focused classification (without the MD fitting done in MDSPACE) result in most particles not useful for inferring distinct and visible N-terminal domain conformations

- THe methodological details reported in 3.4 are more complete than what reported in the methods section

- What is the method used to generate the relative free energy landscape of Fig. 4A?

- Line 388: how is the clustering analysis done in MDSPACE? Which parameters?

- Line 392: it seems that Fig. 4A reports a free energy landscape obtained by Bolzmann inverting bidimensional (from UMAP reduction) histograms. This has to be reported in the methods section

- Line 408: how is the multi-dimensional local minima calculation done? Do the authors intend structural topological clusters from particles comparison? From Fig. 4A (on first two dimensions) it is not clear the existence of the declared multiple local minima. There seems to be a global minimum with roughly (8;4)  first two components

- Lines 430-432:"Indeed, one particular conformational state could be repeated 6 times in the data (n x 60° rotations around the C6 symmetry axis, n = 6), which increases even more the complexity of the embedding" This is true, and the actual n° of conformations are indeed an artificial underestimation. Anyway, the heterogenity and incomplete variability in up-/down- of  the predicted classes suggest that is more a sampling problem rather than a software limitation. Indeed, the free energy profile that was estimated in Fig. 4A is not clearly presented: what are the more common conformations? Why different clusters pass over regions of the free energy landscape that have a different probability? Are clusters structurally homogeneous or the landscape is biased by the dimensionality reduction used to represent it (so structurally differences at N-terminal domain are masked)?

- Lines 439-445:"Thus, the rigid-body alignment of the fitted asymmetrical models with the C6-symmetrical model may not result in the perfectly aligned copies of the states that only differ in the position of the N-domain in the ring, which induces additional difficulties for the UMAP to separate the truly different states." Not clear, as UMAP is not doing any clustering, but a dimensionality reduction. Does it mean that initial alignment errors may cause errors after fitting to a degree to which UMAP multi-dimensional representations are biased to stay in between two "possible clusters", making impossibile for the clustering algorithm  used (which one?) to separate them?

- Paragraph 3.5 is very clear. Please remove the many methodological details from the previous result pargraphs, in order to improve readability

- Lines 481-483:"This analysis aimed to ascertain any potential coordination or synchronicity in the movements observed among the N domains within the context of the hexameric assembly" There could be allosteric communication between subunits even without synchronous movements of the N-terminal domain. Have you tried to predict any signal of allosteric communication between domains from the full ensemble?

- Lines 483-496: this part is clear, but the categorization you did seems just a manual bias: from Fig. 6A it seems that there is a continuum angular variation between -30 to +30, without discrete states. So, your discritization is approximate and arbitrary, and this has to be stated. Otherwise you should show that is the result of a structural clustering of conformations in a space with > 2 dimensions

- Figure 6. Please replace "conformations" with "conformation index", as it is easy to get confounded with the "n° of conformations"

- Lines 509-550: this text is a repetition of the introduction. Please simplify in two sentences integrated with the rest of the discussion from line 551

- Line 555:"...the N-domains remained were poorly defined..." Fix

- Lines 581-584:"Upon analyzing variations at the monomer level, we observed a trajectory involving up and down rotations of the N-domain within a 60° range. The range of conformations analyzed at the monomeric level showcased a continuous spectrum of conformations, revealing a swing of 60° in the N-domains around a central position" The previous two statements are equal in contents

- Line 591:"Approximately 27% of the monomers had 5 domains in similar positions" Not clear, how could a monomer have 5 domains, if each monomer is made of 3 domains?

Comments on the Quality of English Language

Just a minor check

Author Response

We appreciate the time and effort you have taken to provide us with your feedback and an extended list of editorial edits. We have addressed your comments and included your suggested edits as described in detail below.

Major comment: “it is not possible to actually define discrete conformations (as it is evident in Fig. 6), so the authors provided an arbitrary classification to estimate the statistical preference of certain conformations in the hexamer.”

The reviewer accurately pointed out that discrete clusters (discrete conformations) cannot be clearly separated in the continuous conformational landscapes (Fig. 5A, and 6A). It is important to note that the first UMAP component aligns with the principal axis of motion of the N-domains in the monomers. The three regions labeled as -30°, 0°, and +30° in Fig. 5A are subregions that contain particles most similar to the average position (group 0°) and those most different (groups -30° and +30°). The numerical values indicate the approximate rotation angle of the N-domain in each specific sub-region compared to the average N-up conformation. Regarding Fig 6A, the landscape was discretized by segmenting it into three regions along the first UMAP component to ensure an equal number of monomer particles across each region. These three regions encapsulate modeled structures characterized by a rotation range of the N-domain spanning from -30° to -10°, -10° to +10°, and +10° to +30° with respect to the average N-up conformation (now labelled as groups -20°, 0° and 20° in Fig. 6A). Therefore, the classification used to assess the statistical preference of the monomer’s conformations in the hexamer is not arbitrary. It relies on the data-driven process of discretizing the first UMAP component into regions with an equal number of points, making it inherently driven by the data rather than predetermined geometric considerations.

We have clarified these points in the revised manuscript as follow:

- Results section, lines 424-435: “Considering a continuous nature of the obtained conformational landscape (Figure 5A) ... the monomeric N domain in the p97 up-conformation (PDB 5FTN).”

-  Methods section, lines 646-651: “Then, the obtained conformations of the monomers were mapped back ... the corresponding monomer in the hexameric p97.”

Minor comments:

- Line 107: "...iteratively match the deformed model to particles..." The particles that are intended are the 2D Cryo-EM images?

Correct, point clarified in the text by replacing particles with “2D Cryo-EM images” (now lines 127-128).

- Line 129: the rotation was previously described as known [5,9]. What is the difference detected by the authors with MDSPACE? It seems to be the big rotation for the N-terminal stabilized mutant; anyway, this aspect is not underlined or valorized in the manuscript.

The rotation previously described for the N domains was between two conformations, one with the N-domain co-planar to the D1 ring and the other with the N-domain 14Å above the D1 ring (which also corresponds to a rotation of ~75° of this domain).  In this manuscript, we analysed the conformational flexibility of the so-called “up-conformation” and showed that it is not one conformation but a continuum of conformations.

To emphasize this point, we have modified and added information in the last chapter of the introduction, lines 149-154: “Moreover, MDSPACE at the monomeric level … spanning from 45 to 105° compared to the coplanar conformation.”

We also added one sentence in the results section, lines 400-402: “This analysis shows that the N-domains of the mutant are … from 45 to 105° compared to the coplanar conformation”.

- Line 231: Which is the input vector to UMAP? RMSD distance between all atoms (which atoms? alpha Carbon?), torsion angles? Inter-residue distances?
UMAP is not a clustering technique. What is the clustering technique used?

The input to UMAP is a set of Cartesian coordinates of Cα atoms. This was now clarified lines 627-630: “Uniform Manifold Approximation and Projection (UMAP) [84] ... onto a low-dimensional conformational space.”

The clustering in the UMAP-based low-dimensional conformational landscape of the hexamers and monomers in Fig. 4A and Fig. 5A, respectively, was performed manually by considering the density of points in different regions of the landscape (interactive grouping of points in dense regions). The clustering in the UMAP-based landscape of the monomers in Fig. 6A was performed by discretizing it along the first UMAP component (which corresponds to the principal axis of motion) to obtain three regions with the same number of particles.

This is now clarified as follows:

In the Material and Methods section, lines 634-637: “The clustering in UMAP-based... grouping of pints in dense regions”; lines 643-646 “The UMAP dimension reduction ... grouping of points in dense regions)”; lines 647-649 “To this goal  ... three regions with the same number of particles”.

In the Results section, lines 323-324: “These two clusters ... in the two-dimensional space”; lines 441-444: “Six examples of such sub-clusters ... each containing approximatively 10,000 particles”; Lines 390-392 ”By selecting regions along this trajectory ... several positions of the N-domains”; Lines 425-428 Therefore, we decided to discretize  ... each region containing the same number of particles”.

 - Lines 260-262:"these densities also had a fuzzier appearance when compared to the N-domains in the projections of the PDB structures with the up and down N-domain conformations (Figure S1B,C)." Not clear

This sentence was modified (now lines 171-174): “We observed that the densities … clearer densities observed in the D1 and D2 domains (Figure S1)”.

- Lines 289-292:"The local resolution of the C1 map indicates higher resolution in the central densities corresponding to the D1 and D2 rings compared to the N-domains (Figure 1F–G). The low local resolution in the N-domain regions is coherent with their observed appearance/disappearance in the EM map and may suggest a strong flexibility of the N-domains." This is redundant.

We consider that this explanation is required for non-expert reader and kept this section.    

- Paragraph 3.2 can be simplified, as repeats many methodological steps previously reported

This section (now section 2.2) has been simplified in the revised manuscript as to avoid duplication, in particular information about the mask design has been removed, lines 298-301 original version of the manuscript.

- Line 310: "Regrouping particles from the classes where the N-domain was resolved enabled us to improve the resolution of the monomeric map from 3.9 Å to 3.0 Å " In this group the N-terminal domain is visible on monomers? How many resampled/subtracted images belong to the first two classes in which the N-terminal domain is solved? This is important to understand the degree of uncertainity in the classification and how much continous is the conformational shift

Yes, in this group the N-terminal domain is visible on monomers (as shown in Figure 2). The number of images per class is also indicated in figure 2. 166,158 particle images ( 115,727+50,431) belong to the first two classes (or ~10.1% of the images).

The information has been added in the Results section, lines 225-228 “In the second round of classification, the N-domains were visible in only two classes…  ~10.1 % of all monomer images)”.

The fact that no densities were visible, or very poorly defined, in the other reconstructions is consistent with the continuous flexibility of the N-domain in this population. We have clarified this lines 246-249: “However, this approach did not enable the segregation … averaging out the N-domain densities.”

- Line 328: why only "class 5" molecules? Are they better resolved? Because it was known they are up-conformation?

The rationale for choosing this class has been added to the manuscript, lines 263-264: “as this class contained the majority of the images and showed reasonable well-defined N-domains”, 

- Lines 358-359:"However, this approach did not comprehensively capture all potential conformations of these domains." I would say the main point is that using a monomer focused classification (without the MD fitting done in MDSPACE) result in most particles not useful for inferring distinct and visible N-terminal domain conformations

Thank you for the suggestion. We agree with your conclusion. However, at that stage in the manuscript, the MDSPACE has yet been presented.

We have added this explanation, lines 295-297: ”The monomer focused classification on its own … visible N-terminal domain conformations.”  

And clarified that point in the discussion, lines 662-665: “In the case of this study… visible N-terminal domain conformations”.

 - The methodological details reported in 3.4 are more complete than what reported in the methods section.

The redundant information in the results section has been removed (lines 368-372 “Because MDSPACE performs one simulation ….  ensuring accurate simulation of native dynamics”) and relevant information added to the Material and Methods section (lines 610-619 “NMMD includes MD simulations, which were performed using … while ensuring accurate simulation of native dynamics”; lines 623-627: “The remaining fitted structures… for the NMMD fitting in MDSPACE”; Lines 634-657: “The clustering in the UMAP-based…  being the Boltzmann constant and T the temperature of the system”.   

- What is the method used to generate the relative free energy landscape of Fig. 4A?

The density of the points in the low-dimensional space obtained by UMAP is converted into free energy differences through the Boltzmann factor Δ G⁄k_B  T=-ln⁡(n⁄n_0 ) by counting the number of particles n in each region of the space and the number of particles in the most populated region n_0, with k_B being the Boltzmann constant and the temperature of the system.

This is now explained in the Methods section (lines 652-657).

- Line 388: how is the clustering analysis done in MDSPACE? Which parameters?

This question seems redundant with their previous inquiry about line 231. In the revised manuscript, we provided all the information necessary for understanding how the clustering was performed in the low-dimensional space obtained by UMAP (please see our answer above and our changes to the Material and methods section described above).

- Line 392: it seems that Fig. 4A reports a free energy landscape obtained by Bolzmann inverting bidimensional (from UMAP reduction) histograms. This has to be reported in the methods section.

As explained above, we added the following to the Materials and Methods section (lines 652-657) to describe the method used to obtain the free energy landscape: “The density of the points in the low-dimensional space obtained by UMAP is converted into free energy differences .. and  the temperature of the system. “

- Line 408: how is the multi-dimensional local minima calculation done? Do the authors intend structural topological clusters from particles comparison? From Fig. 4A (on first two dimensions) it is not clear the existence of the declared multiple local minima. There seems to be a global minimum with roughly (8;4)  first two components.

The computed UMAP space had a total of ten dimensions as two dimensions were not sufficient to describe the conformational variability of the N domains in p97 hexamers. In Fig 4A, we show only the two first components as it is not possible to display a ten-dimensional space. However, the “subclusters” in this figure were manually obtained by considering all ten UMAP dimensions. In this ten-dimensional space, multiple local minima were observed. The subclusters shown in Fig 4A are projections of the sub-clusters from the ten-dimensional space onto a two-dimensional space. Fig 4A shows 6 examples (arbitrary number) of the local energy minima regions in the ten-dimensional space. It shows a diversity of the conformations of the N domain.

This was clarified in lines 343-349: “The first two UMAP components allow to … encircled in Figure 4A”.

- Lines 430-432:"Indeed, one particular conformational state could be repeated 6 times in the data (n x 60° rotations around the C6 symmetry axis, n = 6), which increases even more the complexity of the embedding" This is true, and the actual n° of conformations are indeed an artificial underestimation. Anyway, the heterogenity and incomplete variability in up-/down- of  the predicted classes suggest that is more a sampling problem rather than a software limitation. Indeed, the free energy profile that was estimated in Fig. 4A is not clearly presented: what are the more common conformations? Why different clusters pass over regions of the free energy landscape that have a different probability? Are clusters structurally homogeneous or the landscape is biased by the dimensionality reduction used to represent it (so structurally differences at N-terminal domain are masked)?

During our work, we first attempted the exploration of higher order UMAP components at the hexameric level, hoping to identify more homogeneous clusters. Due to the complexity of the problem, we concluded that it would be more efficient to switch to the analysis at the monomeric level, instead of continuing with the landscape analysis at the hexameric level. This is the reason why we are not showing more common conformations in the landscape at the hexameric level but, rather, a diversity of conformations (Fig. 4). After the analysis at the monomeric level, we can conclude that the main reason for not having perfectly homogeneous clusters is continuous flexibility resulting in a continuous conformational landscape (Figs 5A, Fig. 6A), in which case clearly separated clusters (discrete homogeneous classes) are not possible to obtain.

- Lines 439-445:"Thus, the rigid-body alignment of the fitted asymmetrical models with the C6-symmetrical model may not result in the perfectly aligned copies of the states that only differ in the position of the N-domain in the ring, which induces additional difficulties for the UMAP to separate the truly different states." Not clear, as UMAP is not doing any clustering, but a dimensionality reduction. Does it mean that initial alignment errors may cause errors after fitting to a degree to which UMAP multi-dimensional representations are biased to stay in between two "possible clusters", making impossible for the clustering algorithm  used (which one?) to separate them?

We are sorry about this confusion. We moved some sentences from the Results section to the Materials and Methods section dedicated to MDSPACE, which we believe better clarify the need for the rigid-body alignment of the NMMD fitted structures before the use of UMAP. Indeed, UMAP is a dimension reduction method and we want it to focus only on the conformational heterogeneity (not on the heterogeneity that combines conformational and pose heterogeneities). Thus, the NMMD fitted structures were rigid-body aligned to discard their rigid-body motions induced during MD simulations (during the NMMD fitting in MDSPACE), as described in the original MDSPACE publication.

Lines 623-630: “The remaining fitted structures … the initial conformation for the NMMD fitting in MDSPACE.”

- Paragraph 3.5 is very clear. Please remove the many methodological details from the previous result paragraphs, in order to improve readability.

Thank you. All methodological details were moved to M&M section.

- Lines 481-483:"This analysis aimed to ascertain any potential coordination or synchronicity in the movements observed among the N domains within the context of the hexameric assembly" There could be allosteric communication between subunits even without synchronous movements of the N-terminal domain. Have you tried to predict any signal of allosteric communication between domains from the full ensemble?

This would require further work and could be attempted in the future.

- Lines 483-496: this part is clear, but the categorization you did seems just a manual bias: from Fig. 6A it seems that there is a continuum angular variation between -30 to +30, without discrete states. So, your discritization is approximate and arbitrary, and this has to be stated. Otherwise you should show that is the result of a structural clustering of conformations in a space with > 2 dimensions.

We explained our discretization method in the Results section, lines 424-435: “Considering a continuous nature of the obtained conformational landscape ... in the p97 up-conformation.”

and in the Material and Methods section, lines 646-650: “Then, the obtained conformations of the monomers were mapped back … corresponding monomer in the hexameric p97.”

- Figure 6. Please replace "conformations" with "conformation index", as it is easy to get confounded with the "n° of conformations"

Fig 6 was modified accordingly.

- Lines 509-550: this text is a repetition of the introduction. Please simplify in two sentences integrated with the rest of the discussion from line 551

This text has been shortened, in particular, lines 509-515, lines 520-526 and lines 529-530 from the original manuscript have been deleted.

- Line 555:"...the N-domains remained were poorly defined..."

Changed to “the N-domains were poorly defined” (now line 472).

- Lines 581-584:"Upon analyzing variations at the monomer level, we observed a trajectory involving up and down rotations of the N-domain within a 60° range. The range of conformations analyzed at the monomeric level showcased a continuous spectrum of conformations, revealing a swing of 60° in the N-domains around a central position" The previous two statements are equal in contents

Text modified to “The analysis of the variations at the monomer level using MDSPACE showcased the continuous spectrum of conformations, revealing a swing of 60° in the N-domains around a central position” (lines 498-500) .

- Line 591:"Approximately 27% of the monomers had 5 domains in similar positions" Not clear, how could a monomer have 5 domains, if each monomer is made of 3 domains?

Thank for noticing this type-o. Text should have read “approximately 27% of the hexamers had 5 N-domains in similar positions” (line 507).

Reviewer 2 Report

Comments and Suggestions for Authors

The authors analyzed the conformational variability of the hexameric AAA+ ATPase p97, a complex with a 6-fold rotational symmetric core surrounded by six flexible N-domains. They applied a newly developed method, MDSPACE, and detected a novel conformation adopted by only 2% of the particles. The authors do not provide the biological significance of this conformation. An explanation should be provided if this conformation is not an artifact of the sample preparation.  The work has been carefully designed and all sections (Introduction, Results, Discussion) are well-written and informative.

Author Response

We thank this reviewer for their succinct and positive review. The biological relevance of the conformation adopted by 2% of the particles was discussed lines 597-613 (now 566-581). As indicated, the analysis of other cryo-EM data sets obtained with other mutants and the wild-type p97 protein in the presence and absence of co-factors is required to assess the biological significance of this conformation (lines 526-527).

In the revised manuscript, we added that single molecule fluorescence studies would also help assessing the dynamic properties and the tendency of p97 to adopt this D1-D2 ring open conformation, lines 528-530: “Validation through single molecule fluorescence studies ... ring open conformation.”